# The human metabolome and machine learning improves predictions of the post-mortem interval

Rasmus Magnusson [1] ✉, Carl Söderberg [2], Liam J. Ward [2,3], Jenny Arpe [1], Fredrik C. Kugelberg [2,3], Albert Elmsjö[2,3], Henrik Green [2,4,5] & Elin Nyman [1,5]

An accurate prediction of the time since death, known as the post-mortem interval, remains a critical research question in forensic and police investigations. Current methods, such as rectal temperature and vitreous potassium levels, only provide reliable post-mortem interval estimations up to 1–3 days. In this study, we use metabolomic data from routine toxicological screenings using femoral whole blood samples ($n$=4876 individuals) with known post-mortem interval of 1–67 days. We develop a neural network model that predicts the post-mortem interval with a mean/median absolute error of 1.45/1.03 days in unseen test cases, outperforming six other machine learning architectures. Pseudo-time series clustering of important model features reveals distinct metabolite dynamics, including markers of lipid degradation, mitochondrial dysfunction, and proteolysis. To assess generalizability, we apply the trained model to independent test data ($n$ = 512 individuals) collected in a different year and analyzed on a separate mass spectrometry platform. Despite cross-platform variability, the model retains predictive performance (mean/median absolute error 1.78/1.29 days). We further show that robust models can be trained using only a few hundred cases, supporting scalability. Our findings demonstrate that post-mortem metabolomics, even when derived from routine toxicological workflows, can enable accurate post-mortem interval predictions and may offer a transferable framework for future forensic applications.

Accurately determining the time of death or the time elapsed since a person died, also known as the post-mortem interval (PMI), is of critical importance in forensic and police investigations. In general casework, knowledge of PMI can significantly improve the understanding of the circumstances surrounding the death of the deceased. An accurate estimate of PMI in a homicide case significantly impacts the ongoing investigation and the ability to reconstruct the chain of events that lead to death.

Despite considerable research[1–7], current best practices are unable to provide accurate and reliable estimates of PMI in a variety of scenarios. Typically, one or several of the following indicators are used to estimate PMI of a corpse: livor mortis (discoloration of the skin), rigor mortis (stiffening of the body), algor mortis (body adjustment to surrounding temperature), and quantitative measurements of potassium concentration in vitreous humor. The accuracy of these methods varies. Generally, the evaluation of rigor mortis and livor mortis is

[1]Department of Biomedical Engineering, Linköping University, Linköping, Sweden. [2]Department of Forensic Genetics and Forensic Toxicology, National Board of Forensic Medicine, Linköping, Sweden. [3]Department of Biomedical and Clinical Sciences, Linköping University, Linköping, Sweden. [4]Department of Biomedical and Clinical Sciences, Science for Life Laboratory, Linköping University, Linköping, Sweden. [5]These authors contributed equally: Henrik Green, Elin Nyman. ✉e-mail: rasmus.magnusson@liu.se

subjective and can vary significantly between individuals[4]. The rectal body temperature, on the other hand, can be seen as the current gold standard method for PMI estimation[3] This method is effective until the body reaches the surrounding temperature, which typically occurs within 24 hours post-mortem[3]. The potassium concentration in the vitreous humor of the eye is also reliable in the early post-mortem time period since the concentration increases proportionally with PMI for 0-48 hours after death[8]. However, a notable increase in the uncertainty of this PMI estimate occurs at longer PMIs[8].

Beyond early post-mortem time periods, methods are typically less precise and mostly cover substantially later PMIs. Factors that can be used to predict PMI include, but are not limited to, insect activity[3], body decomposition[3,9,10] and examination of skeletal remains[11–13]. Indeed, there is a need for more precise methods for PMI determination that can be applied beyond the first days post-mortem.

The human metabolome, which includes all endogenous substances of low molecular weight, has shown promising potential in PMI estimations when analyzing biochemical changes in body fluids and/or body tissues after death[14–20]. PMI has been identified as the main factor driving post-mortem metabolomic changes in various soft tissues and fluids[16]. Although studies on human samples with a controlled PMI are scarce, there are some notable studies that relate PMI to metabolomic changes. For example, studies have reported a correlation between PMI and the levels of metabolites such as hypoxanthine, choline, creatine, betaine, glutamate, and glycine in serum, vitreous humor, and aqueous humor[17] and threonine, tyrosine and lysine in muscle tissue[20]. In addition, taurine, glutamate, and aspartate in vitreous humor have also been associated with PMI[21]. However, there is still a lack of large-scale human metabolomic studies capable of generating predictive models for PMI that would be suitable for forensic and police investigations.

Machine learning has emerged as a useful tool for PMI prediction in different contexts. For example, a study employing a support vector regression model trained on 39 metabolites from 36 rats reported a mean squared error of 10 hours when predicting PMIs up to 72 hours on independent test data[14]. Multi-omics data from human bone samples have also been used to estimate PMIs in the range of 200–800 days, using methods such as principal component analysis and partial least squares discriminant analysis[22]. Furthermore, changes in human gene expression have been successfully used to predict PMIs within the 0–30 hour range using gradient boosted trees[23]. Despite these promising advances, current approaches have not yet been developed or validated using large-scale forensic case material, limiting their generalizability and practical applicability in forensic settings.

In this study, we address these gaps using post-mortem metabolomics data from 4876 authentic forensic cases, representing the largest human metabolomics dataset analyzed for PMI prediction to date. Using femoral whole blood samples and high-resolution mass spectrometry data originally generated for toxicological screening, we developed a neural network model that accurately predicts PMI with a mean absolute error (MAE) of 1.45 days and a median absolute error of 1.03 days. To assess generalizability, we tested the model on independent data ($n = 512$) measured on a different high-resolution mass spectrometry platform and collected during a separate time period. Despite known challenges with platform differences, the model retained predictive capability, with a MAE of 1.78. Furthermore, we explored various of-the-shelf supervised machine learning methods, finding that four of the six models demonstrated significant predictive power, although none surpassed the performance of the neural network model. Our findings demonstrate that the human post-mortem metabolome contains a robust and informative signal for PMI estimation and that routine toxicology data can be repurposed to develop predictive models, to suggest the potential for post-mortem metabolomics in the prediction of PMI, pending further validation in different settings and conditions.

## Results

### A neural network model fills the gap of reliable PMI estimations in critical time frames

We used metabolomics data from 4876 femoral blood samples from forensic investigation, with 2305 selected features (see Methods). For these samples, PMI values ranged from 1 to 67 days. 97% of PMI values were between 1 and 13 days, with the median PMI 5 days. We have therefore chosen to show the model performance for PMI 1–13 days in the main manuscript and across all recorded PMIs in the supplementary Fig. 1. We randomly divided the samples into training, validation, and test sets in proportions 80% (3907 profiles), 10% (471 profiles), and 10% (471 profiles), respectively (Fig. 1a). The histogram for the PMI-distributions of data is found in Supplementary Fig. 2.

Given the number of samples, we chose to apply a feed forward neural network (FFNN) regression model to predict PMI. We first performed a hyperparameter optimization with respect to the model performance on the validation data. In detail, we tested 30 FFNN models with 1–4 hidden layers, each containing 32 to 512 hidden nodes. In addition, each layer had a dropout rate of between 0.05 and 0.5. The model always terminated in an additional single output node, and all activation functions used the rectified linear unit (ReLU). The models were implemented with a learning rate between 1e-4 and 1e-2. Furthermore, we explored whether incorporating feature selection as the initial step would improve model performance by implementing a custom-built attention mechanism as the first layer in the optimization process. The model selection included an early stopping with respect to the validation error, with a patience of 25 epochs.

We observed a noticeable robustness in model performance with respect to hyperparameter design. All tested model hyperparameter settings and their respective performances can be found in Supplementary Data 1. Interestingly, we observed a clear trend where the performance was clearly stratified by the implementation of the custom-built attention layer. The 18 best-performing models all had such an attention layer, whereas the nine worst-performing models did not. The overall best-performing model contained an instance of the attention layer, a dense layer with 288 hidden nodes with a dropout rate of 0.15, and an output layer. The optimal learning rate for this model was set to 0.005.

Having selected and applied the FFNN regression model to the test data and compared the predicted PMI values with the actual PMI values, we found the model to accurately predict PMI values, with a mean absolute error of 1.45 days (Fig. 1b). The median absolute error was 1.03 days. Given that some of the actual PMI values had a resolution of 48 hours ($n = 985$), with a typical uncertainty of less than 24 h, the relative error of the predictions was comparable to the underlying uncertainty of the training data.

To benchmark our approach, we compared the FFNN model with potassium-based PMI estimation using an established model[8]. We assumed conditions similar to our study population (ambient temperature 6 °C and the age of the deceased being 40 years) and estimated PMI with a 95% prediction interval. For potassium concentrations reflecting the earlier post-mortem period (e.g. 10 and 15 mM/L), the model[8] provides PMI estimations of 0.66–2.05 days and 1.16–4.39 days, respectively. For higher concentrations (e.g. 20 nM/L), the model[8] provides PMI estimations of 1.49–8.24 days, which is not useful in forensic practice. In comparison, our FFNN model produced estimated PMIs of 2.5–5.9 days at an actual PMI of 3 days and 4.2–8.3 days at an actual PMI of 7 days (95% prediction intervals). These intervals should not be confused with the MAE, which summarizes the average error across all predictions.

### The model generalized to independent test data

To evaluate the generalizability of our approach, we collected independent test data ($n = 512$) from a different time-period and analyzed on a separate high-resolution mass spectrometry platform. By jointly

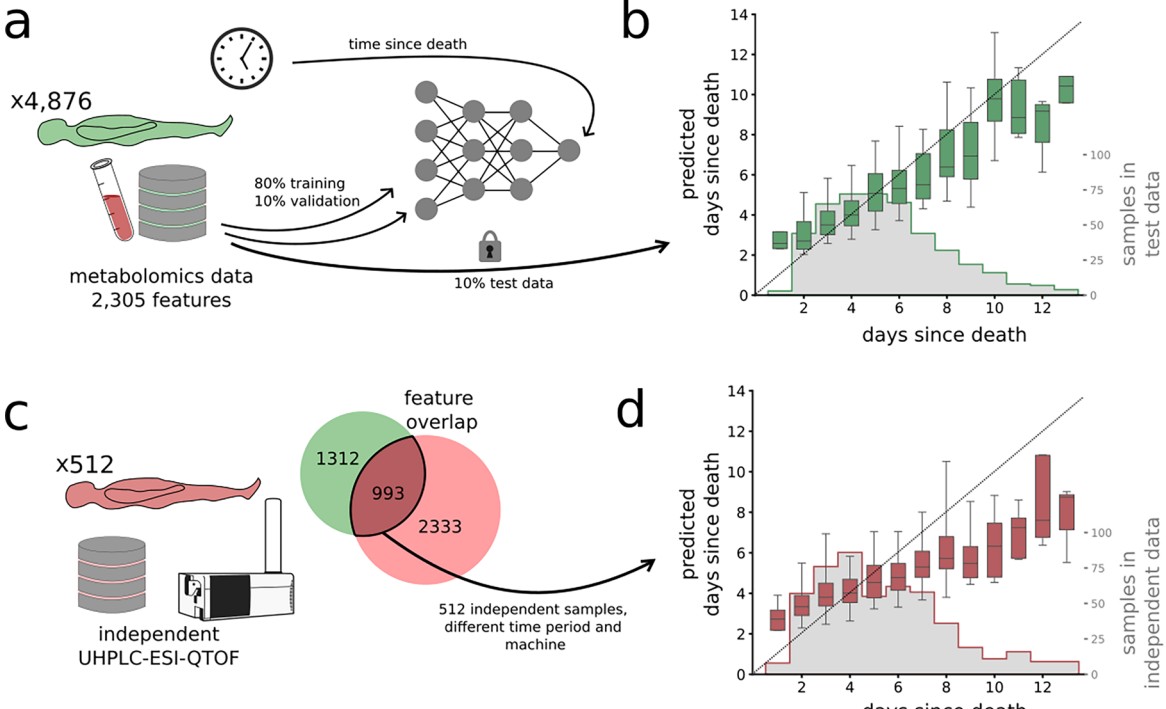

**Fig. 1 | Study design and model performance. a** We used our compendium of 4,876 post-mortem metabolomic profiles to train a feed forward neural network (FFNN) regression model to predict the post-mortem interval (PMI). We trained the model with the best performing hyper-parameters using early-stopping with respect to the validation error. **b** We found a mean absolute error of 1.45 days when predicting PMI in the test data. We show predictions for PMI 1–13 days ($n = 455$) since 97% of the samples fell within this range. For predictions for the whole PMI range (1–67 days), see Fig. S2. The distribution of test data (grey histogram) was in line with that of the training and validation data (Fig. S3). Shown are the distributions of model estimates as a function of the actual PMI. Box plots show the median (centre line), interquartile range (25th–75th percentiles; box), and whiskers extending to the 2.5th and 97.5th percentiles; outliers are not shown. **c** Using 512 independent test samples from another instrument and time period, we predicted the corresponding PMIs using the 993 overlapping features. **d** The model predicted PMI of the independent test data with a MAE of 1.78, without any form of retraining. We show predictions for PMI 1–13 days (n=502). Again, shown are the distributions of model estimates as a function of the actual PMI. The boxes show the median (center line), interquartile range (25th–75th percentiles; box), and whiskers extending to the 2.5th and 97.5th percentiles. Outliers are not shown. Source data are provided as a Source Data file. Created in BioRender. Green, H. (2026) https://BioRender.com/usy2kin.

analyzing the spectra from both datasets, we identified 993 overlapping and 2333 exclusive features in the new dataset (Fig. 1c). We applied the trained FFNN model to the independent test data. Despite known challenges associated with platform differences, the model retained predictive capability when used on the independent test data (MAE = 1.78 days, Fig. 1d). The MAE was thus comparable to that observed in the original dataset, and the median absolute error was 1.29. We also evaluated model performance by sex using the same independent test data, since the original data contained 72% male samples. The mean error for the female samples was approximately 5% higher than for male samples; however, this difference was not statistically significant (two-sided Mann-Whitney U test, $p = 0.26$).

### A compendium of supervised machine learning methods shows significant information embedded in the data

Neural networks are known for their ability to learn from complex, non-linear patterns in data, and for their requirement of large datasets to do so. We continued the study by analyzing whether less complex alternative machine learning models could also accurately predict PMI. To this end, we built a compendium of models and fitted them to the training data. The model types included in this compendium were a ridge regressor, a least absolute shrinkage and selection operator (LASSO), a support vector regression (SVR), a random forest regressor, a gradient boosting regressor, and a K-nearest neighbor regressor (k-NN). We again trained these models using 80% of the data (3,907 profiles) and tested them using 10% of the data (471 profiles). We observed clear predictive power across five of the six models (Fig. 2a),

although the FFNN had the lowest mean absolute error (Fig. 2b). The differences between the models' performances were all statistically significant (two-sided Wilcoxon signed-rank test $p < 0.05$) except I) the LASSO vs the SVR, II) the LASSO vs the Random Forest, III) the LASSO vs the Gradient Boosting, and IV) the SVR vs the Gradient Boosting. All models achieved $R^2$ values substantially above zero, indicating predictive performance beyond random guessing. In particular, the feedforward neural network (FFNN) reached an $R^2$ of 0.56, followed by LASSO (0.49), Gradient Boosting (0.46), Random Forest (0.40), and SVR (0.39) (Fig. 2c). K-NN and Ridge performed weaker, with $R^2$ values of 0.18 and 0.13, respectively.

### Pseudo-time series clustering reveals biologically expected metabolomic changes

Having found that all tested machine learning methods satisfactory predicted the time since death from the metabolomic profile, we sought to characterize these post-mortem changes. To this end, we extracted the metabolite features used by the FFNN by selecting the metabolite features for which the activation of the input variable was significantly correlated with the recorded PMI, using a Benjamini-Hochberg corrected $p$-value of a Spearman's rank correlation. For these variables ($n = 810$), we created pseudo-time series by averaging the abundance of each metabolite feature over individuals with the same PMI. We performed a hierarchical clustering to extract pseudo-time-related groups of metabolites and found three groups of post-mortem changes (Fig. 3). This approach revealed a cluster of 158 metabolite features with decreasing abundance over time, a cluster of

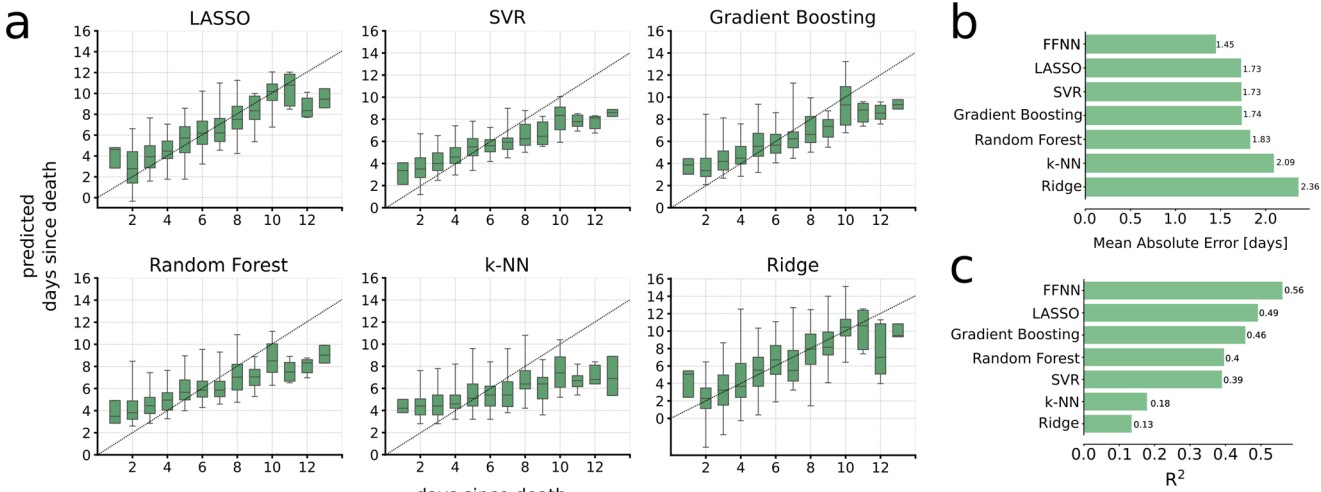

**Fig. 2 | The comparative performance of alternative machine learning methods. a** We applied a compendium of six supervised regression models to the same metabolomic profiles used to train the FFNN. By evaluating all models on the unused validation data ($n = 455$ individuals), we found an ability to predict PMI across all methods. Shown are the distributions of the respective model estimates as a function of the actual PMI. The boxes show the median on the center line and cover the 25–75th percentiles, with whiskers extending to the 2.5th and 97.5th percentiles. Outliers are not shown We found the FFNN to outperform these regression models, both in terms of **b** the MAE and **c** $R^2$. Source data are provided as a Source Data file.

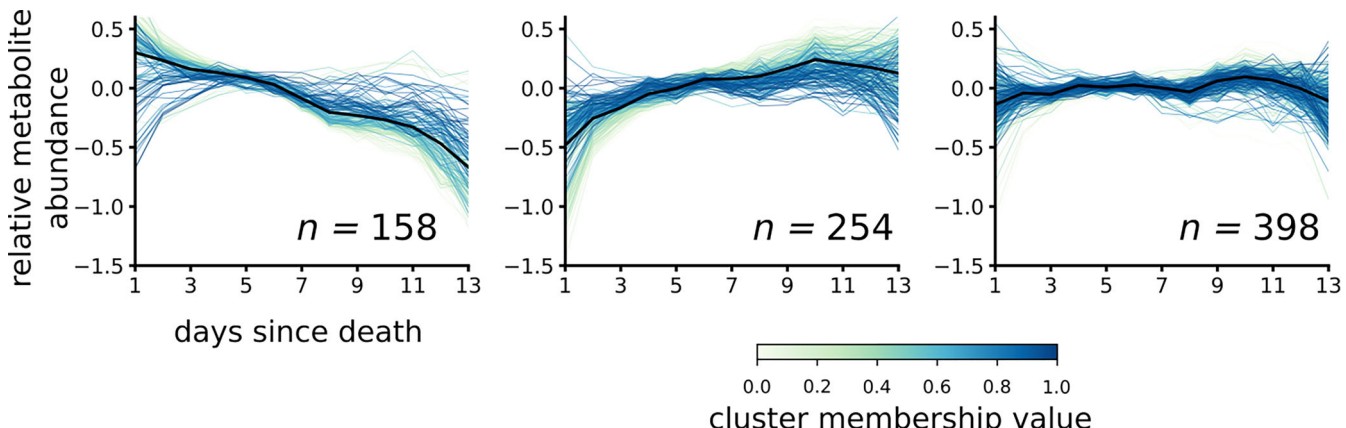

**Fig. 3 | Clustered pseudo-time series reveal broad trends in post-mortem metabolite changes.** We computed the pseudo-time series of the metabolites features of the attention layer of the FFMM model that correlated with PMI and divided them into three groups using hierarchical clustering. We found a set with a downward trend containing 158 metabolites, one with an upward trend (254 metabolites), and one with a fluctuating abundance (398 metabolites).

254 metabolite features where the abundance increased, and a cluster of 398 metabolite features where the abundance showed more complex patterns over the pseudo-time scale.

From the visualization of the pseudo-time series in clusters, the more dispersed signal at early ( < 3 days) and late ( > 11 days) PMI becomes visible (Fig. 3). This aligns with the model's less accurate ability to predict low and high PMI (Fig. 1d). While it is not surprising that longer PMIs introduce more uncertainty (e.g. due to surrounding temperatures and other external factors), we wanted to investigate the predictive power of the models at the earlier time points. Therefore, we analyzed the respective predictions of the test data at PMI = 1. We found that all tested models overestimated the actual PMI at day 1, with overestimates ranging from 1.79 (FFNN) to 3.46 (K-NN) days and a median overestimate of 2.56 days (Supplementary Fig. 3).

Metabolomic features putatively identified by mass-to-charge matching during functional analysis (MetaboAnalyst) and/or database matching (HMDB) are presented in accordance with cluster membership and relationship to PMI pseudo-time. For features decreasing in relation to PMI pseudo-time, 38 features were identified to 24 unique metabolites (Supplementary Data 2). Of these, acylcarnitines ($n = 12$)

and lysophosphatidylcholines ($n = 11$) were the most prevalent. For features increasing in relation to PMI pseudo-time 43 features were identified to 30 unique metabolites (Supplementary Data 3). Here, amino acids and dipeptides were the most prevalent. For features fluctuating over PMI pseudo-time, 38 features were identified to 32 unique metabolites (Supplementary Data 4).

**A few hundred samples can be sufficient to predict PMI**

Next, we asked how many samples a forensic institute would need to train a well-performing model. To answer this, we analyzed how the number of samples/metabolomic profiles in the training dataset influenced model performance. We chose to evaluate the predictive error of a LASSO regression model as a function of training sample size. Specifically, we randomly selected $n$ metabolomic profiles from the training data, with $n=[16, 32, …, 1024]$ 150 times for each $n$, and performed a 5-fold cross-validation for model selection. The LASSO regularization parameter $\lambda$ was chosen to minimize the prediction error in a 5-fold cross-validation. We evaluated model performance and found that as few as 256 or 512 samples were sufficient to give indications of PMI predictions, achieving an average MAE of 2.05 and

# a    prediction benchmark

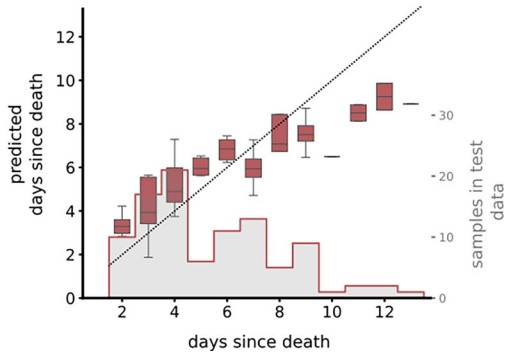

# b    independent benchmark validation

**Fig. 4 | Predictive error as a function of the number of metabolomic profiles used as training data. a** We randomly sampled n metabolomic profiles from the training data, as shown on the x-axis, to train LASSO regression models. The y-axis shows the range of the mean absolute error (MAE) of the corresponding 150 models in each set of profiles. The whiskers show the 95% distribution of the respective prediction errors. We found the mean absolute error of this prediction to decrease towards the error of the LASSO regression model where all 3907 profiles in the training dataset were used (dashed-dotted line). Shown is also the performance of the full FFNN model on the test dataset (dashed line). **b** Using the independent data, now divided into 80% training and 20% testing, we confirmed the finding that a LASSO combined with even a limited amount of data could be used to predict PMI. This new LASSO regression model achieved in independent samples, observing a MAE of 1.49 using the 20% left-out test data. We show predictions for PMI 1–13 days (n=98). Source data are provided as a Source Data file. For both plots, boxes indicate the interquartile range with the median shown as a central line, and whiskers denote the 2.5th and 97.5th percentiles. Outliers are not shown.

1.91 days, respectively (Fig. 4a). As expected, the prediction error decreased consistently as the number of profiles in the training data increased, underscoring how additional data improved accuracy.

Lastly, we sought to test this observation on the 512 independent test data samples. We randomly divided the data into 80% training and 20% test data and trained a LASSO regression model. The sparsity parameter was optimized using a cross-validation on the training data. This model was able to accurately predict PMI (Fig. 4b), with a MAE/median absolute error of 1.49/1.18 days, showing both the predictive power of human metabolomic data on PMI, and again that also limited number of samples can be used to train predictive PMI models. The distribution of all data is found in Supplementary Fig. 2.

## Discussion

We have demonstrated that metabolomics data from authentic forensic cases, combined with machine learning, can predict PMI with relevant accuracy. Using high-resolution mass spectrometry data, originally generated for toxicological screening, we developed an FFNN model with good accuracy (MAE = 1.45 days, Fig. 1). This error is comparable to the uncertainty in the reference PMI values, which in our dataset ranged from a few hours to 48 h, depending on whether the time of death was recorded as certain or probable (see Methods). This inherent uncertainty is particularly problematic for short PMIs, where even small deviations can significantly affect model evaluation. Given the current resolution of data, accurate predictions for PMI=1 are not feasible. A unique strength of this study is the cross-platform validation of the FFNN model. Despite known challenges associated with cross-platform variability, the model retained predictive capability (MAE = 1.78 days), underscoring the translatability of our approach.

Model performance was generally strong but less accurate at the extremes of the PMI range, with a tendency to overpredict at short intervals ( < 3 days) and underpredict at longer intervals ( > 11 days). This pattern was consistent across all tested models (Fig. 2), including k-NN, which is typically less prone to regression-to-the-mean effects, which likely reflects the distribution of training data, which was concentrated in the mid-range, highlighting the need for more balanced data. While linear models such as SVR and LASSO also performed well, the FFNN outperformed them, which could be due to mild non-linear

dependencies between the metabolomic changes and PMI. The robustness of predictions across different machine learning methods suggests that the post-mortem metabolome contains a broad and informative signal. Future efforts to improve PMI estimation will likely benefit from knowledge-driven modeling strategies, such as[24] that can accommodate complex, multi-layered biological data.

We further explored the biological underpinnings of the model by clustering metabolite features based on pseudo-time dynamics (Fig. 3). For identified metabolites features, the observed decrease in lysophosphatidylcholines (LysoPC) suggests phospholipid membrane breakdown and loss of cell membrane integrity. Likewise, for acylcarnitines the decrease likely reflects loss of enzymatic activity for fatty acid β-oxidation in the mitochondria. Decreases in acylcarnitines have been observed previously in human intoxication cases[25–27] and found indicative of hypoxia-related mechanisms in rat models[28,29]. Therefore, prolonged hypoxia associated with respiratory depression during death may be the link to the observed alterations in acylcarnitines. The increase in amino acids and dipeptides over PMI pseudo-time are indicative of proteolysis and tissue breakdown. This observation supports previous findings in rat models, which show increasing amino acids levels in the early PMI period[20,30]. Together, these metabolite trends over PMI pseudo-time highlight molecular decay processes after death; lipid membrane degradation, mitochondrial deterioration, and proteolysis. These findings align with a recent metabolomics study showing time-dependent shifts in acylcarnitines and amino acids in human post-mortem samples[18]. Some metabolites change both with PMI and with cause of death. For example, LysoPC(14:0) and LysoPC(15:0) decreased over PMI pseudo-time but have also been identified as discriminative markers for cause of death[27]. This highlights the need for future models to integrate both PMI and cause-of-death information. Similarly, the incorporation of sex and other metadata could potentially be incorporated to future modeling approaches.

Although our dataset is uniquely large, we show that robust models can be trained on substantially smaller sample sizes of a few hundred samples (Fig. 4), with error decreasing approximately linearly with increasing sample size. We further tested the robustness of the approach using the collected independent data (n = 512) and found that a LASSO regression model achieved good predictions also in this

case (MAE = 1.49 days). This demonstrates that the metabolomics signal is highly informative across two different sets of data and that even forensic institutes with relatively few cases can leverage this approach effectively. This scalability supports the feasibility of implementing PMI prediction models.

PMI estimation remains challenging due to biological variability and environmental factors that affect all available methods. Our findings indicate that metabolomics-based models can provide narrower prediction intervals than traditional potassium-based approaches beyond the first days. Alternative strategies, such as microbiome-based analyses[31], have achieved sub-day resolution under sterile, controlled conditions in animal models[32], but their accuracy decreases under more realistic conditions (e.g., soil exposure), with MAE from 3 days using different microbiota for the predictions[6]. Unlike these controlled studies, our work is based on real forensic case material, introducing greater variability but reflecting operational conditions.

A key strength of our work is demonstrating that data originally generated in toxicological screenings can be repurposed for metabolomics-based PMI estimation, enabling the development of prediction models without additional analytical cost. As more forensic toxicology laboratories adopt high resolution mass spectrometry, the potential for similar applications will increase. We acknowledge that forensic institutes may follow different protocols and may not routinely collect comparable data, which limits direct transferability. Another important limitation of our study is the lack of detailed documentation on transport and storage conditions, which prevents calculation of accumulated degree days (ADDs). Since temperature strongly affects decomposition, ADDs would offer a more precise measure than PMI. However, our cohort reflects real-world forensic cases with standardized handling, with cooling typically initiated within 48 hours after death. Despite this important limitation, the model performed well, suggesting that the metabolomic signal is robust under operational conditions.

Looking ahead, further validation of our approach in diverse forensic settings is essential to ensure the generalizability. Ongoing collaborations with international forensic institutes will be crucial in collecting data under varied environmental conditions and storage practices. Another priority is the development of user-friendly software tools, enabling integration of metabolomics-based PMI estimation into routine workflows. Ultimately, our goal is to establish a standardized, reliable method for PMI estimation that could, following further research and validations, be widely implemented, improving the accuracy and efficiency of forensic investigations worldwide.

## Methods

### Statistics and reproducibility
The study was approved by the Swedish Ethical Review Authority (Dnr 2019-04530 and Dnr 2025-0249-02). Due to the retrospective nature of the study, the need of informed consent was waived by Swedish Ethical Review Authority. All methods were carried out in accordance with relevant guidelines and regulations, and all samples were irreversibly anonymized prior to analysis. Autopsy cases were obtained from the Swedish National Board of Forensic Medicine. After transportation to the autopsy site, bodies were stored in controlled indoor environments, typically in morgue refrigeration units, which substantially reduces variability in decomposition rate compared to outdoor or uncontrolled conditions. For more details, see Supplementary Methods 1.

For the original study, we included all autopsy cases admitted between 2017-09-01 and 2019-03-14. For the independent test data, cases were collected between 2021-01-01 and 2021-12-31. The inclusion criteria were as follows: availability of femoral blood, age ≥18 years, and completion of toxicological screening using high-resolution mass spectrometry. For more experimental details, see Supplementary Methods 1. No statistical method was used to predetermine sample

size. No data were excluded from the analyses; The experiments were not randomized; The Investigators were not blinded to allocation during experiments and outcome assessment.

From the original study period (2017-09-01 to 2019-03-14), all autopsy cases with a certain or probable death date (as described below) were included (in total 4876 autopsy cases). The most frequent causes of death in the studied population, each with more than 100 documented cases, included complications of cardiovascular disease ($n = 748$), acute poisoning with one or more drugs ($n = 659$), hanging ($n = 572$), alcohol poisoning (221), drowning (194), trauma resulting in multiple internal and external injuries (150), and gunshot wounds (119). The demographic characteristics of the selected cohort can be summarized as follows: age, median = 56 years (interquartile range = 39–69); sex, male = 3544 (72.3%); BMI, median = 25.7 (interquartile range = 22.5–29.6); PMI, median = 5 days (interquartile range 4-7).

For the independent test data (2021-01-01 and 2021-12-31), we randomly selected 512 autopsy cases with a certain or probable death date (as described below).

### Calculating the post-mortem interval
In the database of the National Board of Forensic Medicine in Sweden, the date of death can be ascribed and coded in two different ways: certain and uncertain. For the uncertain cases, a probable date of death is indicated together with the date the deceased was last seen alive. The present study includes all cases in which the date of death is certain ($n = 3954$). Additionally, cases with a probable date of death are included if the date when the person was last seen alive is the day before the body was found ($n = 922$). This approach allows for the inclusion of cases where, for example, a person was last seen alive in the evening and found dead the following morning. However, this also gives an uncertainty of up to 48 hours in extreme cases (body last seen day 1 at 00:01, body found day 2 at 23:59).

In some cases, the date of death was ascribed as probable even if the date last seen alive was the same date as when the deceased was found dead. In Swedish forensic practice this is sometimes done to indicate that the death was unwitnessed (i.e., last seen alive in the morning and found dead in the evening).

In the present study, PMI is defined as the time (in days) between date of death, certain or probable as described above, and the date of the autopsy in which sampling was performed. Autopsies are typically performed during morning office hours in Sweden. Therefore, e.g. PMI = 2 can for the cases with a probable death date represent between 32 hours (body last seen and found 23:59 day 1, autopsy in the morning day 3) and 80 hours (body found day 2, last seen day 1 at 00.01, autopsy in the morning day 4).

### Software implementation
The post-mortem metabolomic data from the samples were pre-processed in R using the xcms package and CAMERA package, as in ref. 33. This workflow included peak detection, retention time alignment, and feature grouping, and gap filling using the XCMS fillPeaks algorithm, resulting in 2305 metabolomic features. Missing values occurred when a metabolite peak was not detected in a given sample during XCMS processing; these missing peak intensities were imputed as zero values, reflecting metabolite abundance below limit of detection.

The computational analyses were performed in Python v. 3.12.4, mainly relying on the packages Keras v. 3.3.3, scikit-learn v. 1.5.0, numpy v. 1.26.4, scipy v. 1.13.1, matplotlib v. 3.8.4, and seaborn v. 0.13.2.

### Data processing
We normalized the data using a log-transform (Eq. 1).

$$x_{norm} = \ln(x + 1) \tag{1}$$

where $x$ is the peak intensity and $x_{norm}$ is the log-transformed expression. The log transformation was applied to stabilize variance and reduce skewness, thereby mitigating heteroskedasticity inherent in the raw data. Following the log-transformation, we standardized the values of each metabolite using a z-transform (subtracting the mean and dividing by the standard deviation of each log-transformed metabolomic profile). This step ensured that all metabolites were centered at zero and scaled to unit variance, facilitating comparability across variables in downstream machine learning models. The log transformation and z-transformation addressed different aspects of the data distribution: the former reducing skewness and heteroskedasticity, and the latter ensuring comparability of variables by placing them on a common scale. Exploratory analyses of metadata suggested minimal batch effects or other sample-to-sample variation, and therefore no additional correction was applied.

We normalized and standardized the dataset prior to splitting. Given the large sample size, the differences in distribution between the full dataset and the training subset are minimal, and the impact on model performance is expected to be negligible. We randomly divided the dataset into training (80%), validation (10%), and test (10%) using probabilistic assignment.

### Data harmonization between the different datasets
To harmonize the variables of the independent test data to that of the original mass spectrometry dataset, we aligned the features between the original and new datasets using a peak mapping approach. As such, each feature in the original dataset, defined by its median mass-to-charge ratio (m/z) and retention time (RT), was matched to candidate features in the new dataset that fell within the corresponding m/z and RT bounds. Among these candidates, the feature with the minimal squared distance from the original feature's m/z and RT was selected as the best match. If there was no such candidate, the feature was set to 0. This mapping allowed the new dataset to be reordered to match the feature structure expected by the model.

### Neural network design and training
To predict PMI, we implemented a feed forward neural network regression model using the Keras package. To determine the optimal design of the model, we initiated a hyper-parameter optimization to select the number of hidden layers, the number of hidden nodes in each layer, and the dropout rate for each layer. Additionally, we implemented the option to train the model with a custom attention layer as an input layer such that each metabolite would be individually passed to a single node. The rationale behind this was that such an attention layer would serve as a data-driven feature selection algorithm.

We trained each hyper-parameter setting three times and selected the best performance using an early stopping algorithm with a patience of 25 epochs with respect to prediction error on the validation data. The early stopping algorithm was also used to restore the best weights during training. All tested hyperparameter sets can be found in the Supplementary Data 5.

### Training of alternative machine learning models
We trained a compendium of alternative machine learning regression models using the same training data as for the FFNN model. These models included the two linear regression models, Ridge and LASSO, where the respective $L_2$ and $L_1$ penalty weights were selected using the Scikit-Learn built-in RidgeCV and LassoCV implementations. We also implemented a gradient boosting and a random forest regression model, each with the default 100 estimators, as per the default setting on the Scikit-Learn package in Python. Furthermore, we also implemented a K-nearest neighbor (K-NN) regressor and a support vector regression (SVR), both as implemented in Scikit-Learn. We trained the models using the assigned training data, and, for consistency, tested the

models on the test data. The used settings of the alternative machine learning methods can be found in the Supplementary Data 5.

### Extracting learned model structures and feature identification
We sought to extract which metabolomic features were used in the decision-making processes of the respective models. While such an extraction of input-output dependencies is non-trivial for neural networks[34,35], we utilized the attention layer of the FFNN to estimate the usage of input variables. By analyzing these input node activations and correlating them with PMI, we generated a list of metabolomic features ranked by importance.

All metabolomics features included in the FFNN were uploaded to MetaboAnalyst (version 6.0) for functional analysis, which is suitable for untargeted metabolomics data, relying on the assumption that putative annotation at the compound level can collectively predict group level functional changes defined by set of pathways of metabolites[36].

Metabolomic features significant for decision-making in the FFNN were putatively identified and annotated by reviewing the compound hits from the functional analysis in MetaboAnalyst, and/or by matching the mass-to-charge ratio (m/z; ± 5 ppm) to the online human metabolomic database (HMDB; https://hmdb.ca).

### Reporting summary
Further information on research design is available in the Nature Portfolio Reporting Summary linked to this article.

## Data availability
We have used the online human metabolomic database (HMDB; https://hmdb.ca) to extract features from our data. The metabolomics data generated in this study have been deposited at Figshare (https://doi.org/10.6084/m9.figshare.30931784) and also together with the relevant code on GitLab at https://gitlab.liu.se/rasma87/pmi_metabolomic_prediction. Furthermore, source data are present. Source data are provided with this paper.

## Code availability
The samples were pre-processed in R using the xcms package and CAMERA package. The computational analyses were performed in Python v. 3.12.4, mainly relying on the packages Keras v. 3.3.3, scikit-learn v. 1.5.0, numpy v. 1.26.4, scipy v. 1.13.1, matplotlib v. 3.8.4, and seaborn v. 0.13.2. The code for the model-based analysis is available at our GitLab page at https://gitlab.liu.se/rasma87/pmi_metabolomic_prediction which contains instructions on how to reproduce our results.

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

## Acknowledgements

The study was supported by the Swedish Research Council (grant no.: Dnr 2023-01407 Green, Dnr 2019-03767 Nyman), the Swedish Fund for Research Without Animal Experiments (grant no.: S2021-0008, F2022-02 Nyman), and the Strategic Research Area in Forensic Sciences at Linköping University (Magnusson, Ward).

## Author contributions

R.M., C.S., L.J.W., F.C.K., A.E., H.G., and E.N. conceived and planned the experiments. C.S., L.J.W. and A.E. collected and preprocessed the data. R.M., L.J.W., and J.A. performed calculation and modeling analyses with input from all the authors. H.G. and E.N. supervised the work. All authors discussed the results and contributed to the final manuscript.

## Funding

## Competing interests

The authors declare no competing interests.
