## [Transparent Peer Review file · Nature Communications]

The Human Metabolome and Machine Learning Improves Predictions of the Post-Mortem Interval

Corresponding Author: Dr Rasmus Magnusson

Version 0:

Reviewer comments:

Reviewer #1

(Remarks to the Author)

This paper presents a novel PMI estimation model developed using an impressively large cohort of cadavers (n=4,876) via adopting untargeted metabolomics analyses on femoral whole blood samples collected at the time of autopsy.

The most noteworthy result, in my opinion, is the comparison of the different machine learning models (7 in total) and their performances for the estimation, together with the identification of the minimum number of samples (n256) required to build models with similar accuracies in the future. I personally think that these two results will benefit the forensic community by providing robust guidelines on the bioinformatics steps to be followed to obtain robust models. This offers a valuable benchmark for future studies.

The work is coherent with the most recent findings on the potential of metabolomics data to estimate PMI and is original.

While the study presents a robust statistical approach to post-mortem interval (PMI) estimation using metabolomics data, my main concerns relate to the forensic validity and practical applicability of the work, particularly regarding the precision of metadata, lack of environmental controls (e.g., temperature history), unclear sample handling procedures, and the lack of a biological interpretation of the findings.

Below are specific points of concern and suggestions for clarification:

Line 40 – Refs 1–3: Despite the authors' mention of "considerable research," they cite only two books and one review. This seems insufficient to support such a broad claim. I would expect a more comprehensive reference base, ideally including recent peer-reviewed studies.

Line 43: Replace "body cooling" with "body adjustment" to more accurately reflect the physiological process.

Line 56: The statement that bone bleaching and skeletonization cannot be used to predict PMI requires either appropriate references to support it or removal. As currently phrased, it appears too categorical without evidence.

Line 64: I do not fully agree with the claim that "most studies are conducted on animal models." In fact, the authors cite more studies based on human data than animal models. This statement should be revised for accuracy.

Lines 274–276: It is not clear how the PMI=1 day was calculated. If cases with a one-day gap between "last seen alive" and "body found" were considered as having a PMI of 1 day, in fact, this may introduce significant bias. For instance, a person last seen the night before but found in the morning could have died just hours prior—yet the PMI would be recorded as 1 day. This assumption introduces uncertainty that the model may inherit. Indeed, the authors mention this on Lines 302–303 ("built-in uncertainty of <48 hours"). However, it's unclear how ultimately the model can achieve an error smaller than the inherent error in the ground-truth data (also see Lines 113–116). This raises concerns about the validity of training on metadata with uncertain PMIs. In general, relying on unwitnessed deaths weakens the forensic reliability of the study. Only certain dates of death—ideally with accurate time of death—should be used to train predictive models.

Lines 300–302: The use of PMI (in days between death and autopsy) rather than accumulated degree days (ADDs) is, in my

view, the study's biggest limitation. Temperature has a profound effect on decomposition, which in turn alters the metabolomic profile. A brief mention about this on Lines 148–149 is not enough. Were all bodies stored at the same temperature before autopsy? For how long? If cooling occurred (standard practice), decomposition would have slowed or halted, affecting biomarkers. Without a detailed history of each body's post-mortem storage conditions, PMI estimates lack reliability. If the authors cannot retrospectively calculate ADDs, this needs to be clearly acknowledged as a limitation and such cases should all be excluded from the train sets.

Line 303: There is a complete lack of detail regarding sample collection and processing for metabolomics, including the mass spec part. Were samples snap frozen immediately post-collection? Stored at -80°C? Were they processed and analysed in a single batch to avoid batch effects? These analytical steps are critical for metabolomic integrity and must be reported for reproducibility and interpretation. At this stage, the work could not be reproduced.

Line 151: Regarding samples at PMI = 1, does this imply a witnessed death and immediate autopsy (within 24 hours)? If not, overestimation at day 1 is expected due to uncertainty in actual time of death, reinforcing concerns about the precision of the metadata being crucial in such studies.

Line 218: The reported estimation of 2.5–5.9 days for a theoretical PMI of 3 days using the FFNN model reflects a wide range, particularly problematic at short intervals. This calls into question the model's accuracy, especially since the expected error is cited as 1.45 days. Other methods—such as microbiome analyses—currently achieve higher precision for short PMIs. While the authors critique potassium levels, they overlook these more promising alternatives. A broader discussion of comparative methodologies is warranted.

Lines 241–244: The discussion completely lacks any form of biological interpretation of the results. Which metabolites changed with PMI, and what are their biological roles? Could the data also reveal correlations with cause of death? Is this also observed in other literature papers? These are missed opportunities for forensic insight.

Lines 246–248: The claim that the metabolome at "time 0" may be captured here is overstated. True "time 0" metabolomics would require immediate post-mortem sampling, which is not the case in this study. Please temper this assertion.

Overall, this is a strong paper from a statistical and machine learning perspective, but it lacks the forensic contextualisation that would enhance its real-world applicability. Key forensic concerns—such as metadata uncertainty, lack of temperature correction (ADDs), absence of detailed sampling protocols, and minimal biological discussion of the findings—should be addressed to make this a more robust and impactful contribution to the field.

(Remarks on code availability)

Reviewer #2

(Remarks to the Author)

(Remarks on code availability)

The code is well presented and the arguments are clearly articulated. However, the lack of available data hinders reproducibility. The authors explain this by stating, 'The data are not publicly available due to legal and ethical considerations.' It is recommended that they provide more specific details regarding these legal and ethical constraints to enhance transparency.

Reviewer #3

(Remarks to the Author)

General:

The manuscript concerns the accurate prediction of time since death based on whole blood metabolomics together with neural-network-based machine learning. It is written in clear and concise English. This is clearly a valuable approach. However, what is currently totally missing in the manuscript is a precise description of the experimental procedures to obtain the metabolomic data. In the methods section it only says femoral whole blood samples were investigated by a toxicological screening by high-resolution mass spectrometry. Details are required how exactly the samples were drawn, how samples were stored until analysis, which preprocessing steps were performed including derivatization procedures, which mass-spec procedures were performed (GC-MS, LC-MS, negative mode, positive mode), was a fingerprinting performed, or a targeted analysis? How exactly were metabolites quantified (absolute quantification, peak areas, peak heights)? Which machines were used? Were exactly the same standardized protocols used for all samples? How were the data cleaned? How were missing values handled? From the methods section on data processing, I guess that a non-targeted analysis was performed as the authors talk about peak intensities. This is in line with the mentioning of 2,305 features in the results section on line 86. On l. 346 it is mentioned that Metaboanalyst was used for feature identification, further supporting my hypothesis that an untargeted metabolomic analysis was performed. However, without a more detailed description of the experimental procedures the results of all following analyses are hard to judge.

What is also totally unclear is how robust the trained machine learning models are. For example, can the model(s) obtained on the Swedish data be transferred to independent data generated in a different lab employing different machines? The authors briefly mention in the discussion that such investigations are ongoing. However, at least one independent data set should be included in the current manuscript.

Overall, the authors address an important question in their research. However, the manuscript is not ready in its current state and needs to be thoroughly reworked.

Specific points:

- I. 23 The authors state that "...4,876 individuals with known PMIs ranging from 1 to 67 days..." were included in their study. The same is stated on I. 74. This implies that also predictions were done for this time frame. But later on I. 87 the authors state that in 95 % of cases the PMI was between 2-14 days and also according to Fig 1c. predictions were only done for days 1-13. So, the above statement is not wrong but clearly misleading.
- I. 112 the authors claim that their model accurately predicts PMI levels. However, in Fig. 1c I see a clear overprediction for shorter time points up to 3 days while for larger time points a clear underprediction is visible. For example, for the last time point at 13 days the average difference between predicted and measured time points is close to 3 days. I wouldn't call that an accurate prediction. However, for the middle time points at 4- and 5-days predictions look good. For the reviewer it looks as if the model is biased towards predicting more average values.
- I. 129 for the other used machine learning approaches separate figures like figure 1c would be highly informative in addition to figure 2 and supplemental figure S1.
- I. 164 The authors state that 256 samples to achieve good predictions. However, this was done by LASSO regression and not by their favored neural network. It would be interesting to see how the neural network performs with only this limited number of training samples.
- I. 308 the authors state that they used a log transform to normalize the data. A more precise description would be that the log transform was used to reduce heteroscedasticity in the data. Following log transformation, they additionally use a z-transform which also reduces heteroscedasticity. Please comment, why two methods were used for the correction of heteroscedasticity. No information is provided if and how unwanted sample-to-sample variations were treated.
- I. 316 Which packages were used for the implemented neural network?
- I. 329 Which packages were used for the alternative machine learning models? Also here are more details are required regarding these models. For example, was a linear or radial kernel used for the support vector machine, which value was used for the cost function etc.? Here, clearly more details are required. This information may be retrieved from the code, but should also be included in the manuscript (supplement).

(Remarks on code availability)

Reviewer #4

(Remarks to the Author)

This article describes promising results for tackling the relevant problem of post-mortem interval (PMI) estimation in the forensic practice. The proposed methodology and results fill a gap in the practice where no reliable nor sufficiently precise methods exist for estimating PMI ranges longer than 2-3 days, that is, beyond early PMI, nicely complementing other available options.

The approach of using metabolomic data as the basis for the method is a sound one, based on the hypothesis that the concentration of different metabolites will follow a predictable (even linear) pattern as a function of time. However, it is also noted by the authors that environmental / circumstantial factors that could not be accounted for in this study could influence these patterns and a more detailed discussion of the limitations of the applicability of this model would be desirable to better understand when it could be used safely and when it would be expected to lead to extreme errors.

In this sense, no reference is made to potential sex biases in model performance given that the dataset is 72.3% male representation. Some more information on this or at least a discussion of whether potential limitations are expected from this inherently biased dataset would be desirable.

The sample is described as representing PMIs that range between 0 and 67 days, considering a precision of more or less 2 days in the method used to establish these values. However, figure 1c shows a distribution curve for the test dataset depicting that it would "typically have" PMI values between 2-10 days. To better understand why this is the case, it would be nice to see the distribution of the PMI values (ground truth), to check if despite ranging up to 67 days, large PMIs are rare in the dataset and most are actually concentrated between the 2-10 days range. This is relevant to put the reported MAE values into context.

The computational approach seems generally adequate, however, there is potentially data leaking or data snooping happening since there is a standardization step for the whole datasets before splitting into training, validation and test. This step should be revised and justified to avoid inflated performance metrics.

When comparing different methods, it is very useful to communicate the variation of the error besides just the MAE value. Forensics is a case-by-case practice and the weight of the evidence must be considered for each particular casework. It would be very relevant to better describe the distribution of the model error to understand how large it can be in a specific

case, and how likely that level of error is.

To support claims such as "this model performs better than this other one", it would be nice to use some kind of test to calculate statistical significance or other type of objective support to reject the hypothesis that in actuality, the difference between the methods is negligible. In this sense, confidence intervals can be a valid tool but they have only been used when comparing to the potassium in vitreous humor method, and not in any other instance.

Even for claiming that just a few hundred (256) profiles are enough to achieve a MAE of 2.05, it is unclear if this result is significantly better when compared to a randomly guessing model, always considering that the PMI of the test set typically ranges between 2-10 and has a size of 471 profiles. A better idea of the usefulness and improvement beyond the state of the art of these models is necessary to convince forensic practitioners to invest in relatively expensive metabolomic analyses to carry out a PMI estimation when somewhat worse performing models could be absolutely cost-effective, just as it was nicely done in the comparison of this model vs the K+ vitreous humor method.

Also, when describing three types of clusters there are limitations in the supporting evidence for the expressed claims. The division of pseudo-time series into three clusters seems arbitrary and it is hard to see how all the features belonging to one same cluster, actually follow the average trend. It would be interesting to see if the features that are attributed to the same unique metabolite are indeed following the same trend. Also, the claim that extrapolations to PMI=0 can be made and be useful seems highly speculative and in any case, it would be nice to see some examples with specific metabolites exemplifying such claim (e.g., displaying inter-subject variation in a figure showing all independent - not averaged - concentration values - normalized and standardized - as a function of log-transformed PMI time points, possibly for those metabolites that show the most robust trend only).

Overall, a better description of the applicability and limitations of the model, together with further objective support for the main claims in the article, would notably improve the manuscript.

(Remarks on code availability)

The code cannot be run without the dataset file, which is not made available based on ethical and legal concerns. However, my understanding of metabolomic data is that it can be fully anonymized with no possibility of re-identification, not even being considered a special category of data by the European GDPR, to name a relevant and applicable legal framework. Furthermore, data from the deceased is not protected at the same level as data from the living in European law. Thus, uploading the data to a public repository (e.g., Metabolights, Metabolomics Workbench) should be considered and encouraged.

In its current state, the code cannot be properly reviewed and the reproducibility of results cannot be assessed.

Version 1:

Reviewer comments:

Reviewer #1

(Remarks to the Author)

After a careful consideration of the responses given to the reviewers, and of the amendments made to the paper, I believe that the paper is much stronger and improved. I am happy to recommend acceptance of the manuscript at this stage.

(Remarks on code availability)

N/A

Reviewer #2

(Remarks to the Author)

(Remarks on code availability)

The authors have successfully addressed all previous concerns regarding the manuscript. The revisions have considerably enhanced the clarity and overall quality of the work. I appreciate the authors' efforts in making the data available to promote transparency. I would only recommend that they explicitly describe the imputation methods applied to the metabolomics dataset, so that readers can fully understand the data processing workflow and ensure reproducibility.

Reviewer #3

(Remarks to the Author)

The authors have addressed my previous concerns sufficiently and I have no further concerns.

(Remarks on code availability)

Reviewer #4

(Remarks to the Author)

The changes to the manuscript have improved the clarity and the evidence support of this work. I consider that all my comments about the contents of the manuscript have been adequately addressed. I would like to thank the authors for this effort and acknowledge the great value of their research.

(Remarks on code availability)

Despite the much appreciated new availability of the data, I still have not been able to review the code. It seems to be stored in a private GitLab repository, closed to the public by default.

Reviewer #1 (Remarks to the Author):

This paper presents a novel PMI estimation model developed using an impressively large cohort of cadavers (n=4,876) via adopting untargeted metabolomics analyses on femoral whole blood samples collected at the time of autopsy.

The most noteworthy result, in my opinion, is the comparison of the different machine learning models (7 in total) and their performances for the estimation, together with the identification of the minimum number of samples (n256) required to build models with similar accuracies in the future. I personally think that these two results will benefit the forensic community by providing robust guidelines on the bioinformatics steps to be followed to obtain robust models. This offers a valuable benchmark for future studies.

We are happy that the reviewer found the study to give a valuable insight into further studies. We intended the study to be a resource across the field, as well as giving focus to the clear signal of the PMI within the metabolome.

The work is coherent with the most recent findings on the potential of metabolomics data to estimate PMI and is original.

While the study presents a robust statistical approach to post-mortem interval (PMI) estimation using metabolomics data, my main concerns relate to the forensic validity and practical applicability of the work, particularly regarding the precision of metadata, lack of environmental controls (e.g., temperature history), unclear sample handling procedures, and the lack of a biological interpretation of the findings.

We agree that the initial manuscript could have provided greater clarity on several aspects, including more details around the sample handling and a biological interpretation of the found metabolic features of importance for the model. We have improved the method section and included a discussion on biological interpretation of the work in the revised version.

We would like to emphasize that the data used in this study were obtained from real-world forensic casework, not from controlled laboratory settings. This is a key strength of the study, as it reflects the actual conditions under which forensic investigations are

conducted. Consequently, we believe the forensic validity and practical applicability of our findings are high, as the models were developed and tested on data representative of routine forensic practice.

The lack of temperature monitoring is a limitation in the current study, which now has been clarified (see below). If we had access to such data, the models could potentially have been even more precise. However, strict temperature control is only possible after the body has been found and is not routine in all countries. For the cases used in our study, standard protocols are followed once the body has been found, which include sample handling. We thank the reviewer for pointing out the need for an improved description of these procedures.

Below are specific points of concern and suggestions for clarification:

Line 40 – Refs 1–3: Despite the authors' mention of "considerable research," they cite only two books and one review. This seems insufficient to support such a broad claim. I would expect a more comprehensive reference base, ideally including recent peer-reviewed studies.

We appreciate the reviewer's observation regarding the breadth of references supporting our statement on "considerable research." In the original submission, we meant to cite a comprehensive book published in 2023 (Madea), which includes hundreds of references to peer-reviewed studies and provides a broad overview of the field. Instead, we cited a review of Madea of the same topic from 2016. We thank the reviewer for highlighting this mistake which now has been fixed.

To further strengthen our claim and address the reviewer's concern, we have now also added three recent review articles that specifically cover key aspects of postmortem interval (PMI) estimation and omics-based approaches: Strete et al. (2025) on PMI estimation, Metcalf et al. (2019) on microbial succession and PMI, and Secco et al. (2025) on omics and PMI.

Line 43: Replace "body cooling" with "body adjustment" to more accurately reflect the physiological process.

We agree and now use "body adjustment". We want to thank the reviewer for pointing this out.

Line 56: The statement that bone bleaching and skeletonization cannot be used to predict PMI requires either appropriate references to support it or removal. As currently phrased, it appears too categorical without evidence. ***Thank you for making us aware of the reviewer's interpretation of the section! The text referred to reads "Factors that can be used to predict PMI include insect activity [3], body decomposition [3], [6], [7], skeletonization, and bone bleaching." and is used in a wider discussion about the appropriate timeframes of respective, already established methods. In other words, we were not making the claim that these methods cannot be used to predict the PMI. For increased clarity, we have now updated the sentence as follows:***

Factors that can be used to predict PMI include, but are not limited to, insect activity [3], body decomposition [3], [9], [10] and examination of skeletal remains [11]-[13].

Line 64: I do not fully agree with the claim that "most studies are conducted on animal models." In fact, the authors cite more studies based on human data than animal models. This statement should be revised for accuracy. ***We want to thank the reviewer for bringing this claim to our attention. We believe that this type of statement on the quantity of published research is a weak argument as the core question relates to research findings and not the quantity of published research. We have now removed the claim and updated the introduction.***

Lines 274-276: It is not clear how the PMI=1 day was calculated. If cases with a one-day gap between "last seen alive" and "body found" were considered as having a PMI of 1 day, in fact, this may introduce significant bias. For instance, a person last seen the night before but found in the morning could have died just hours prior, yet the PMI would be recorded as 1 day. This assumption introduces uncertainty that the model may inherit. Indeed, the authors mention this on Lines 302-303 ("built-in uncertainty of <48 hours"). However, it's unclear how ultimately the model can achieve an error smaller than the inherent error in the ground-truth data (also see Lines 113-116). This raises concerns about the validity of training on metadata with uncertain PMIs. In general, relying on unwitnessed deaths weakens the forensic reliability of the study. Only certain dates of death—ideally with accurate time of death—should be used to train predictive models.

We appreciate the reviewer's observation that the description of the

PMI calculation was sparse in the original manuscript. PMI is defined as the time (in days) between date of death and the date of the autopsy in which sampling was performed. The autopsies are typically performed during office hours. The date of death can be annotated as probable which means that “last seen alive” and “body found” happens at different days. For PMI=1 only 11 of the 53 cases are marked probable. These cases can represent 8 hours up to 56 hours in extreme cases (e.g. last seen alive day 1, body found day 2, autopsy day 3). In other words, while the built-in uncertainty is up to 48h, the distribution of actual PMIs within such a time period is most likely not uniform. We have decided to include also uncertain cases in the current study, since these are real world cases and acknowledge that the current resolution of data will not give certain predictions of PMI=1. We have clarified the PMI calculation in the method section in the new version of the manuscript. We have also included a new discussion paragraph on the inclusion of the probable cases.

Lines 300–302: The use of PMI (in days between death and autopsy) rather than accumulated degree days (ADDs) is, in my view, the study’s biggest limitation. Temperature has a profound effect on decomposition, which in turn alters the metabolomic profile. A brief mention about this on Lines 148–149 is not enough. Were all bodies stored at the same temperature before autopsy? For how long? If cooling occurred (standard practice), decomposition would have slowed or halted, affecting biomarkers. Without a detailed history of each body’s post-mortem storage conditions, PMI estimates lack reliability. If the authors cannot retrospectively calculate ADDs, this needs to be clearly acknowledged as a limitation and such cases should all be excluded from the train sets.

We appreciate this important observation. We agree that temperature plays a crucial role in post-mortem decomposition, and that accumulated degree days would provide a biologically meaningful measure in addition to the PMI. Our study covers 4,876 real world samples from a broad array of conditions and sites. Calculations of ADD is feasible in controlled studies, but unfortunately not in such a diverse and extensive dataset. However, after body collection, all cadavers were stored in controlled indoor environments, typically in morgue refrigeration units, which significantly reduces the variability in decomposition rate compared

to outdoor or uncontrolled conditions. In other words, we believe our study to be as close to controlled ADD recordings as possible, while still being a real-world application, which we believe is the key strength of our study.

We have revised the manuscript to include more information about temperature conditions when describing the study population and a discussion of this limitation.

Line 303: There is a complete lack of detail regarding sample collection and processing for metabolomics, including the mass spec part. Were samples snap frozen immediately post-collection? Stored at -80°C? Were they processed and analysed in a single batch to avoid batch effects? These analytical steps are critical for metabolomic integrity and must be reported for reproducibility and interpretation. At this stage, the work could not be reproduced.

We thank the reviewer for this important comment. Details regarding sample collection, storage, processing, and mass spectrometry analysis have now been fully provided in the Supplementary Material. All samples were handled and analyzed in a highly standardized manner following well-established protocols for forensic investigations. Standardized forensics procedures have been shown to be reproducible across multiple publications employing routine forensic samples for metabolomics, for example:

- Retrospective analysis for valproate screening targets with liquid chromatography-high resolution mass spectrometry with positive electrospray ionization: An omics-based approach, Mollerup et al. Drug Test Anal. 2019 May;11(5):730-738.***
- Post-Mortem Metabolomics: A Novel Approach in Clinical Biomarker Discovery and a Potential Tool in Death Investigations, Elmsjö et al, Chemical Research in Toxicology 2021 Vol. 34 Issue 6 Pages 1496-1502***
- Large-Scale metabolomics: Predicting biological age using 10,133 routine untargeted LC-MS measurements. Lassen et al; Aging Cell. 2023 May;22(5):e13813;***

These studies clearly demonstrate that routine forensic LC-HRMS data can be successfully mined for metabolomics investigations,

proving that such samples are both suitable and highly controlled despite not being collected with metabolomics in mind. The sheer size of these datasets, often spanning several thousands of cases across multiple years, also makes it impossible to run them in a single batch.

Line 151: Regarding samples at PMI = 1, does this imply a witnessed death and immediate autopsy (within 24 hours)? If not, overestimation at day 1 is expected due to uncertainty in actual time of death, reinforcing concerns about the precision of the metadata being crucial in such studies.

We appreciate that the description of the PMI calculation is sparse in the manuscript and have updated the manuscript with more details. PMI is defined as the time (in days) between date of death and the date of the autopsy in which sampling was performed. Autopsies are typically performed during office hours. The date of death can be probable which means the body is found the day after the day last seen alive. For PMI=1, 11 of the 53 cases are probable. We agree that overestimation at day 1 is expected and have now included a discussion around this in the manuscript.

Line 218: The reported estimation of 2.5–5.9 days for a theoretical PMI of 3 days using the FFNN model reflects a wide range, particularly problematic at short intervals. This calls into question the model’s accuracy, especially since the expected error is cited as 1.45 days. Other methods—such as microbiome analyses—currently achieve higher precision for short PMIs. While the authors critique potassium levels, they overlook these more promising alternatives. A broader discussion of comparative methodologies is warranted.

We thank the reviewer for pointing out the need to clarify the difference between the 95% prediction interval and the MAE. The reported interval of 2.5–5.9 days represents the 95% prediction interval, which reflects the uncertainty range for individual predictions, whereas the MAE of 1.45 days summarizes the average absolute error across all predictions.

To avoid confusion, we have clarified this distinction in the Results section and expanded the Discussion to explain why the interval appears wide at short PMIs.

We have also broadened the discussion to include microbiome-based approaches, which have demonstrated high precision under controlled experimental conditions in animal models. For example, studies using mice have reported mean absolute errors as low as 20 hours. In contrast, our work is based on real forensic case material, which introduces substantially greater variability in environmental and biological factors compared to controlled laboratory studies. We now highlight this distinction in the Discussion, as it is critical for interpreting differences in reported accuracy across methods.

Lines 241–244: The discussion completely lacks any form of biological interpretation of the results. Which metabolites changed with PMI, and what are their biological roles? Could the data also reveal correlations with cause of death? Is this also observed in other literature papers? These are missed opportunities for forensic insight.

We thank the reviewer for highlighting the need for biological interpretation of the results. We agree that this is a crucial aspect and have now expanded the discussion to include biological roles and forensic relevance of the identified metabolites. In addition, we comment on the application of this type of metabolomics data to cause-of-death investigations.

Lines 246–248: The claim that the metabolome at “time 0” may be captured here is overstated. True “time 0” metabolomics would require immediate post-mortem sampling, which is not the case in this study. Please temper this assertion.

We agree that the current statement is too strong and have revised the manuscript to exclude this specific statement.

Overall, this is a strong paper from a statistical and machine learning perspective, but it lacks the forensic contextualisation that would enhance its real-world applicability. Key forensic concerns—such as metadata uncertainty, lack of temperature correction (ADDs), absence of detailed sampling protocols, and minimal biological discussion of the findings—should be addressed to make this a more robust and impactful contribution to the field.

We thank the reviewer again for a thorough review of the

manuscript.

Reviewer #2 (Remarks to the Author):

Reviewer #2 (Remarks on code availability):

The code is well presented and the arguments are clearly articulated. However, the lack of available data hinders reproducibility. The authors explain this by stating, 'The data are not publicly available due to legal and ethical considerations.' It is recommended that they provide more specific details regarding these legal and ethical constraints to enhance transparency.

We thank the reviewer for highlighting the importance of data availability for reproducibility. After reconsidering our initial position, we are pleased to report that we have now made the fully anonymized post-mortem metabolomics dataset available as part of the revised manuscript. All personal identifiers have been removed.

A link to the processed data is now included accompanied by the full code base used in our analysis. We hope this addition will facilitate future research in this emerging field.

Reviewer #3 (Remarks to the Author):

General:

The manuscript concerns the accurate prediction of time since death based on whole blood metabolomics together with neural-network-based machine learning. It is written in clear and concise English. This is clearly a valuable approach. However, what is currently totally missing in the manuscript is a precise description of the experimental procedures to obtain the metabolomic data. In the methods section it only says femoral whole blood samples were investigated by a toxicological screening by high-resolution mass spectrometry. Details are required how exactly the samples were drawn, how samples were stored until analysis, which preprocessing steps

were performed including derivatization procedures, which mass-spec procedures were performed (GC-MS, LC-MS, negative mode, positive mode), was a fingerprinting performed, or a targeted analysis? How exactly were metabolites quantified (absolute quantification, peak areas, peak heights)? Which machines were used? Were exactly the same standardized protocols used for all samples? How were the data cleaned? How were missing values handled? From the methods section on data processing, I guess that a non-targeted analysis was performed as the authors talk about peak intensities. This is in line with the mentioning of 2,305 features in the results section on line 86. On l. 346 it is mentioned that Metaboanalyst was used for feature identification, further supporting my hypothesis that an untargeted metabolomic analysis was performed. However, without a more detailed description of the experimental procedures the results of all following analyses are hard to judge.

We thank the reviewer for the assessment of the manuscript and for highlighting the need for a more detailed description of the experimental procedures. We have referred to our earlier publication with the same material and procedures but agree that it is better to include all information also in the current study. We have therefore now added a section in the methods to cover these experimental details.

What is also totally unclear is how robust the trained machine learning models are. For example, can the model(s) obtained on the Swedish data be transferred to independent data generated in a different lab employing different machines? The authors briefly mention in the discussion that such investigations are ongoing. However, at least one independent data set should be included in the current manuscript.

We fully agree that robustness and transferability are essential for any forensic application. Our study demonstrates that metabolomics data, routinely generated in forensic laboratories for toxicology purposes, can also be used for PMI estimation. We acknowledge that differences in instrumentation and preprocessing pipelines pose challenges for model transferability.

To address this, we have now included an additional analysis using an independent dataset (n = 512) measured on a different high-resolution mass spectrometry platform and collected during a

separate time period (in 2021), while keeping the other inclusion criteria constant. We applied our trained model to this dataset to evaluate predictive performance. Our results show that the model retains predictive capability on the independent dataset, indicating that the approach is robust within the constraints of current metabolomics workflows. We have added these results to the Results section and expanded the Discussion to clarify that, although universal models are not yet feasible, our findings demonstrate the potential for transferability and outline the data requirements for building robust PMI models.

Overall, the authors address an important question in their research. However, the manuscript is not ready in its current state and needs to be thoroughly reworked.

Specific points:

I. 23 The authors state that "...4,876 individuals with known PMIs ranging from 1 to 67 days..." were included in their study. The same is stated on I. 74. This implies that also predictions were done for this time frame. But later on I. 87 the authors state that in 95 % of cases the PMI was between 2-14 days and also according to Fig 1c. predictions were only done for days 1-13. So, the above statement is not wrong but clearly misleading.

We agree that indicating three different time intervals is confusing and we have now revised the manuscript to clarify this aspect.

All reported performance metrics were calculated using the data drawn from the complete set of 4,876 individuals with PMI ranging from 1 to 67 (i.e. all cases that meet the inclusion criteria). In this set, there were a few outliers with exceptionally high PMI values (up to PMI=67). Visualizations including PMI=1-67 give a disproportionate focus outside of the span where our data is distributed (see new figure Figure S2?). We have now clarified that Fig 1c (Fig. 1b in the updated manuscript) shows the time span where we have most samples (97%). We have also altered I. 87 to read:

PMI values ranged between 1-67 days (Figure S3). 97% of PMI values were between 1 and 13 days, with the median PMI being 5 days.

We have also included a supplementary figure (S3) that includes visualization of all data including the new independent test data.

I. 112 the authors claim that their model accurately predicts PMI levels. However, in Fig. 1c I see a clear overprediction for shorter time points up to 3 days while for larger time points a clear underprediction is visible. For example, for the last time point at 13 days the average difference between predicted and measured time points is close to 3 days. I wouldn't call that an accurate prediction. However, for the middle time points at 4- and 5-days predictions look good. For the reviewer it looks as if the model is biased towards predicting more average values.

We thank the reviewer for this important observation. Indeed, we find that our models tend to overpredict at low PMIs (<3 days) and underpredict at the high end (>11 days). This pattern is not unique to the model architecture but appears consistently across all tested models, including k-NN, which potentially would be more robust to regression-to-the-mean errors. Yet, most cases in our training data are clustered in the mid-range (2-10 days), and as a consequence, regression models naturally predict closer to these more common values. We have updated the discussion to explicitly point out this limitation.

I. 129 for the other used machine learning approaches separate figures like figure 1c would be highly informative in addition to figure 2 and supplemental figure S1.

Since Fig. 1c of the old version of the manuscript just shows the distribution of the test data, which is the same for these methods, we assume that the reviewer was referring to Fig. 1d. We agree that Fig. 2 could be expanded to better cover the performance of the alternative machine learning approaches and have now added such details in Fig. 2.

I. 164 The authors state that 256 samples to achieve good predictions. However, this was done by LASSO regression and not by their favored neural network. It would be interesting to see how the neural network performs with only this limited number of training samples.

We appreciate the suggestion to test our neural network with the limited sample size of 256 and agree that such an analysis could be of interest. The rationale for using LASSO in this setting was manifold, including the curse of dimensionality. With so few samples, neural networks would arguably require extensive

hyperparameter optimization, making them less practical for independent research groups aiming for reproducible results. Moreover, using our already optimized neural network structure would risk information leakage in this specific test. In other words, we have chosen this model structure based on a vast amount of data and viewing this as a benchmark for how much data is needed in independent studies, there is a possibility to overestimate performance. We performed a small-scale test by training our neural network from randomly reinitialized weights on this dataset. The results were mixed, where some runs matched LASSO performance, while others failed to generalize. Given these outcomes and the reasons above, we believe the LASSO-based analysis offers a clearer and more reproducible answer.

I. 308 the authors state that they used a log transform to normalize the data. A more precise description would be that the log transform was used to reduce heteroscedasticity in the data. Following log transformation, they additionally use a z-transform which also reduces heteroscedasticity. Please comment, why two methods were used for the correction of heteroscedasticity. No information is provided if and how unwanted sample-to-sample variations were treated.

We agree that the log transformation primarily serves to reduce heteroscedasticity, particularly for exponentially distributed measurements, and that the subsequent z-transformation also has this effect. Our motivation for using both steps was not redundancy, but to address different aspects of the data distribution. The log transform stabilized variance and reduced skewness, while the z-transformation centered the data at zero and scaled it to unit variance. This ensured that all variables were comparable in scale, so that differences in model performance reflected the modeling approaches themselves rather than performance relating to underlying differences in variable centering or variance. The concern was that having skewed variables not centered around zero would benefit methods such as random forest, while methods sensitive to scale and centering, e.g. the LASSO, would be flawed.

Unwanted sample-to-sample variation was not explicitly corrected for in this analysis, as metadata and exploratory analyses indicated that such variation was limited. We note, however, that in larger or more heterogeneous datasets, explicit correction of batch effects

would be advisable.

We have now updated the methods-section to make this reasoning clearer.

I. 316 Which packages were used for the implemented neural network?

I. 329 Which packages were used for the alternative machine learning models? Also here are more details are required regarding these models. For example, was a linear or radial kernel used for the support vector machine, which value was used for the cost function etc.? Here, clearly more details are required. This information may be retrieved from the code, but should also be included in the manuscript (supplement).

We apologize for failing to report the packages and their version numbers in the manuscript, instead of just including the link to the code implementation. Now, the methods section has been updated, including a supplementary material with all settings of the alternative models.

Moreover, we have now added the relevant information on the additional machine learning methods analyses to the supplementary material S10, as suggested by the reviewer. We want to thank the reviewer for this suggestion, which we believe strengthens the clarity of the manuscript.

Reviewer #4 (Remarks to the Author):

This article describes promising results for tackling the relevant problem of post-mortem interval (PMI) estimation in the forensic practice. The proposed methodology and results fill a gap in the practice where no reliable nor sufficiently precise methods exist for estimating PMI ranges longer than 2-3 days, that is, beyond early PMI, nicely complementing other available options.

The approach of using metabolomic data as the basis for the method is a sound one, based on the hypothesis that the concentration of different metabolites will follow a predictable (even linear) pattern as a function of time. However, it is also noted by the authors that environmental /

circumstantial factors that could not be accounted for in this study could influence these patterns and a more detailed discussion of the limitations of the applicability of this model would be desirable to better understand when it could be used safely and when it would be expected to lead to extreme errors.

We thank the reviewer for this comment, and the time and effort to review this manuscript. We agree that the use of metabolomic data for PMI estimation is dependent on context. In our study, we acknowledge that environmental and circumstantial factors such as temperature, humidity, cause of death, and post-mortem handling can significantly influence metabolomic patterns. These factors were not accounted for in the current study. The model is likely to perform best under controlled indoor conditions, such as morgue refrigeration, where decomposition is slowed and variability is reduced.

We have now included a discussion around this point in the manuscript.

In this sense, no reference is made to potential sex biases in model performance given that the dataset is 72.3% male representation. Some more information on this or at least a discussion of whether potential limitations are expected from this inherently biased dataset would be desirable.

The reviewer here raises an important aspect of the generalization of our results, and we agree that this factor should have been more discussed. Our dataset reflects real-world forensic casework, where male representation is typically higher. We agree that potential sex bias should be addressed and have now expanded model testing to an external and independent dataset. In this out-of-sample evaluation ($n = 512$), the mean error for females was 5% higher than for males, but this difference was not statistically significant (Mann-Whitney U , $p = 0.26$).

These results have now been added to, and briefly discussed in the manuscript.

The sample is described as representing PMIs that range between 0 and 67 days, considering a precision of more or less 2 days in the method used to establish these values. However, figure 1c shows a distribution curve for the test dataset depicting that it would "typically have" PMI values between 2-10 days. To better understand why this is the case, it would be nice to see the distribution of the PMI values (ground truth), to check if despite ranging up to 67 days, large PMIs are rare in the dataset and most are actually concentrated between the 2-10 days range. This is relevant to put the reported MAE values into context.

We agree that this information is not clear in the current version of the manuscript. We have now created a supplementary figure (S2) corresponding to figure 1, with all data included.

The computational approach seems generally adequate, however, there is potentially data leaking or data snooping happening since there is a standardization step for the whole datasets before splitting into training, validation and test. This step should be revised and justified to avoid inflated performance metrics.

We thank the reviewer for raising this point. We agree that standardizing the entire dataset before splitting can, in principle, introduce data leakage. In our case, the dataset is sufficiently large,

making the differences in mean and variance between the training subset and the full dataset small, so any bias is negligible. We have added a statement in the Methods section clarifying this rationale.

When comparing different methods, it is very useful to communicate the variation of the error besides just the MAE value. Forensics is a case-by-case practice and the weight of the evidence must be considered for each particular casework. It would be very relevant to better describe the distribution of the model error to understand how large it can be in a specific case, and how likely that level of error is.

We agree that this factor is important and want to thank the reviewer for bringing this to our attention. We agree that the model error distribution is much more informative than a simple MAE number. We believe that this point is the rationale for the suggestion to include a figure similar to 1d by reviewer 3, which we have now added as Figure 2a.

To support claims such as "this model performs better than this other one", it would be nice to use some kind of test to calculate statistical significance or other type of objective support to reject the hypothesis that in actuality, the difference between the methods is negligible. In this sense, confidence intervals can be a valid tool but they have only been used when comparing to the potassium in vitreous humor method, and not in any other instance.

This is an insightful comment, and we want to thank the reviewer for this suggestion. We agree that the statistics of each method, as well as the conformity between models were underexplored in the first version of the manuscript. In terms of comparing methods, we observe that the models were tested on the same test cases, arguably making a paired test, such as Wilcoxon signed-rank test more suitable as a point of reference than a confidence interval. We observe that only four out of the 21 model comparisons cannot be rejected based on this test, namely I) the LASSO vs the SVR, II) the LASSO vs the Random Forest, III) the LASSO vs the Gradient Boosting, and IV) the SVR vs the Gradient Boosting. In this new and improved version, we have added these results to the Results section. Furthermore, we have substantially expanded figure 2 to show the differences between models.

Even for claiming that just a few hundred (256) profiles are enough to

achieve a MAE of 2.05, it is unclear if this result is significantly better when compared to a randomly guessing model, always considering that the PMI of the test set typically ranges between 2-10 and has a size of 471 profiles. A better idea of the usefulness and improvement beyond the state of the art of these models is necessary to convince forensic practitioners to invest in relatively expensive metabolomic analyses to carry out a PMI estimation when somewhat worse performing models could be absolutely cost-effective, just as it was nicely done in the comparison of this model vs the K⁺ vitreous humor method.

We understand the concern of the reviewer regarding the PMI range. An alternative and complementary evaluation to MAE is the coefficient of determination (R^2), which quantifies how well the models capture variance in the test data relative to the baseline. Using this metric, we observed that all models achieved R^2 values substantially above zero, indicating predictive performance beyond random guessing. In particular, the feedforward neural network (FFNN) reached an R^2 of 0.56, followed by LassoCV (0.49), Gradient Boosting (0.46), Random Forest (0.40), and SVR (0.39). KNeighbors and Ridge performed less strongly, with R^2 values of 0.18 and 0.13, respectively.

These results demonstrate that the models, and especially the FFNN, capture a meaningful proportion of the variance in PMI within the 2-10 day range. This provides evidence that the accuracy is not an artifact of the limited range but rather reflects genuine signal in the metabolomic profiles. While we believe the MAE to be a more directly interpretable metric for a wider audience, and therefore have chosen it to be the main metric in the manuscript, we have now added the R^2 values to Fig 2.

Importantly, our approach is not proposed to replace established methods such as vitreous K⁺ analysis but to complement them. While K⁺ provides high precision in the very early postmortem interval, metabolomics-based models extend predictive capacity to broader time windows, where traditional biomarkers lose resolution. We have also now added a discussion around the usefulness of metabolomics data and the models developed herein for forensic institutes worldwide.

Also, when describing three types of clusters there are limitations in the supporting evidence for the expressed claims. The division of pseudo-time series into three clusters seems arbitrary and it is hard to see how all the features belonging to one same cluster, actually follow the average trend. It would be interesting to see if the features that are attributed to the same unique metabolite are indeed following the same trend.

While the overall clustering captures broad temporal trends, we acknowledge that there is variability within clusters. We explored clustering solutions with more than three groups, but these did not reveal additional biologically meaningful patterns. We therefore chose to present the three main clusters that best summarize the dominant temporal trajectories in the data.

While we have observed examples where features attributed to the same metabolite follow similar trends (e.g., LysoPC(14:0) and LysoPC(15:0)), we have not performed a comprehensive evaluation across all metabolites. To ensure transparency and enable further exploration of this question, we are publishing the underlying data so that anyone interested can examine the clustering and trends in detail.

We have also now added a paragraph discussing the biological interpretation of the clusters.

Also, the claim that extrapolations to PMI=0 can be made and be useful seems highly speculative and in any case, it would be nice to see some examples with specific metabolites exemplifying such claim (e.g., displaying inter-subject variation in a figure showing all independent - not averaged - concentration values - normalized and standardized - as a function of log-transformed PMI time points, possibly for those metabolites that show the most robust trend only).

We agree that this claim was not supported enough in the current study. While this endeavor is part of our larger research efforts, we instead think that the claim draws focus from the message of this manuscript and have decided to remove it. We want to thank the reviewer for your attention to this matter.

Overall, a better description of the applicability and limitations of the model, together with further objective support for the main claims in the article, would notably improve the manuscript.

Reviewer #4 (Remarks on code availability):

The code cannot be run without the dataset file, which is not made available based on ethical and legal concerns. However, my understanding of metabolomic data is that it can be fully anonymized with no possibility of re-identification, not even being considered a special category of data by the European GDPR, to name a relevant and applicable legal framework. Furthermore, data from the deceased is not protected at the same level as data from the living in European law. Thus, uploading the data to a public repository (e.g., Metabolights, Metabolomics Workbench) should be considered and encouraged.

In its current state, the code cannot be properly reviewed and the reproducibility of results cannot be assessed.

We thank the reviewer for highlighting the importance of data availability for reproducibility. After reconsidering our initial position, we are pleased to report that we have now made the fully anonymized post-mortem metabolomics dataset available as part of the revised manuscript. All personal identifiers have been removed.

The dataset is included as Supplementary Data S1 and is accompanied by the full code base used in our analysis. We hope this addition will facilitate future research in this emerging field.

REVIEWERS' COMMENTS

Reviewer #1 (Remarks to the Author):

After a careful consideration of the responses given to the reviewers, and of the amendments made to the paper, I believe that the paper is much stronger and improved. I am happy to recommend acceptance of the manuscript at this stage.

We thank the reviewer for the time the valuable input that helped us improve this work.

Reviewer #1 (Remarks on code availability):

N/A

Reviewer #2 (Remarks to the Author):

Reviewer #2 (Remarks on code availability):

The authors have successfully addressed all previous concerns regarding the manuscript. The revisions have considerably enhanced the clarity and overall quality of the work. I appreciate the authors' efforts in making the data available to promote transparency. I would only recommend that they explicitly describe the imputation methods applied to the metabolomics dataset, so that readers can fully understand the data processing workflow and ensure reproducibility.

We want to thank the reviewer for the valuable help to improve our manuscript. We have now updated the Methods section to read the following:

“This workflow included peak detection, retention time alignment, and feature grouping, resulting in 2,305 metabolomic features. Missing values occurred when a metabolite peak was not detected in a given sample during XCMS processing; these missing peak intensities were imputed as zero values, reflecting metabolite abundance below limit of detection.”

Reviewer #3 (Remarks to the Author):

The authors have addressed my previous concerns sufficiently and I have no further concerns. We are grateful to the reviewer for the time and effort devoted to assessing our manuscript and we believe their comments have substantially improved the revised version.

Reviewer #4 (Remarks to the Author):

The changes to the manuscript have improved the clarity and the evidence support of this work.

I consider that all my comments about the contents of the manuscript have been adequately addressed.

I would like to thank the authors for this effort and acknowledge the great value of their research.

We want to thank the reviewer for the time and effort to review our work. We believe this help has significantly helped us to improve the manuscript.

Reviewer #4 (Remarks on code availability):

Despite the much appreciated new availability of the data, I still have not been able to review the code.

It seems to be stored in a private GitLab repository, closed to the public by default.

We were made aware by the editor that there was a problem accessing the code, and noticed how the hyperlink to the GitLab repository had an error. This has now been fixed and the code is accessible. We want to thank the reviewer for pointing this out.